# Resistance to Gemcitabine in Pancreatic Cancer Is Connected to Methylglyoxal Stress and Heat Shock Response

**DOI:** 10.3390/cells12101414

**Published:** 2023-05-17

**Authors:** Rebekah Crake, Imène Gasmi, Jordan Dehaye, Fanny Lardinois, Raphaël Peiffer, Naïma Maloujahmoum, Ferman Agirman, Benjamin Koopmansch, Nicky D’Haene, Oier Azurmendi Senar, Tatjana Arsenijevic, Frédéric Lambert, Olivier Peulen, Jean-Luc Van Laethem, Akeila Bellahcène

**Affiliations:** 1Metastasis Research Laboratory, GIGA-Cancer, GIGA Institute, University of Liège, 4020 Liège, Belgium; 2Department of Human Genetics, Liège University Hospital, 4020 Liège, Belgium; 3Department of Pathology, Hôpital Universitaire de Bruxelles Bordet Erasme l Hospital, Université Libre de Bruxelles, 1000 Brussels, Belgium; 4Laboratory of Experimental Gastroenterology, Medical Faculty, Université Libre de Bruxelles, 1000 Brussels, Belgium; 5Department of Gastroenterology, Hepatology and Digestive Oncology, Hôpital Universitaire de Bruxelles Bordet Erasme Hospital, Université Libre de Bruxelles, 1000 Brussels, Belgium

**Keywords:** oncometabolite, methylglyoxal, glycolysis, therapy resistance, gemcitabine, metformin, aminoguanidine, HSF1, HSP27, HSP90

## Abstract

Pancreatic ductal adenocarcinoma (PDAC) is a fatal disease with poor prognosis. Gemcitabine is the first-line therapy for PDAC, but gemcitabine resistance is a major impediment to achieving satisfactory clinical outcomes. This study investigated whether methylglyoxal (MG), an oncometabolite spontaneously formed as a by-product of glycolysis, notably favors PDAC resistance to gemcitabine. We observed that human PDAC tumors expressing elevated levels of glycolytic enzymes together with high levels of glyoxalase 1 (GLO1), the major MG-detoxifying enzyme, present with a poor prognosis. Next, we showed that glycolysis and subsequent MG stress are triggered in PDAC cells rendered resistant to gemcitabine when compared with parental cells. In fact, acquired resistance, following short and long-term gemcitabine challenges, correlated with the upregulation of GLUT1, LDHA, GLO1, and the accumulation of MG protein adducts. We showed that MG-mediated activation of heat shock response is, at least in part, the molecular mechanism underlying survival in gemcitabine-treated PDAC cells. This novel adverse effect of gemcitabine, i.e., induction of MG stress and HSR activation, is efficiently reversed using potent MG scavengers such as metformin and aminoguanidine. We propose that the MG blockade could be exploited to resensitize resistant PDAC tumors and to improve patient outcomes using gemcitabine therapy.

## 1. Introduction

Pancreatic ductal adenocarcinoma (PDAC) has the worst prognosis of all solid tumors. It is projected to be the second leading cause of cancer-related death before 2030 [1,2]. The present standard of care for PDAC patients involves conventional cytotoxic agents, among which gemcitabine is the current gold standard for treating advanced PDAC [3]. Gemcitabine, a deoxycytidine analogue prodrug, requires cellular uptake and intracellular phosphorylation for activation. Once activated, gemcitabine acts as a potent inhibitor of DNA synthesis. While gemcitabine triggers significant anti-proliferative and pro-apoptotic responses, it has limited long-term efficacy in most PDAC patients. Several mechanisms of intrinsic or acquired resistance hinder the therapeutic effects of gemcitabine [4]. Therefore, in order to propose new treatment options for the large majority of advanced PDAC patients who will face therapy resistance, it has become mandatory to better understand the mechanisms of acquired chemoresistance.

Initial studies aimed at identifying the mechanisms responsible for gemcitabine resistance have mainly focused on the proteins involved in its intracellular availability and activation. The objective of these early investigations was to find markers predicting which patient will gain a clinical benefit from gemcitabine [5]. Other studies concentrated on resistance mechanisms that, although not affecting gemcitabine metabolism per se, impact on the ultimate ability of the cell to undergo apoptosis in response to gemcitabine. For example, it has been shown that increased de novo pyrimidine biosynthesis competes with gemcitabine activity in glycolytic pancreatic cancer cells [6]. Hypoxia and major desmoplasia, frequently found in PDAC tumors, render cancer cells more resistant to gemcitabine-induced apoptosis [7]. More recently, a transcriptomic signature, based on primary cell cultures and patient-derived xenografts, was used to predict adjuvant gemcitabine sensitivity in PDAC [8].

Cumulative evidence indicates that metabolic reprogramming is central to pathogenesis and therapy resistance in cancer [9], particularly in PDAC cells [6,10,11]. Most PDACs harbor mutationally activated KRAS and loss-of-function mutations in tumor suppressor genes (TP53, SMAD4, and CDKN2A) [12]. These oncogenic activator mutations, as well as prevalent hypoxia, act as potent inducers of the glycolytic pathway in cancer [13,14]. Together, hypoxia and glycolysis contribute to tumor progression and resistance to gemcitabine in PDAC [6,11].

In fact, glycolysis also contributes significantly to the spontaneous generation of methylglyoxal (MG) [15], a very reactive dicarbonyl compound that induces the glycation of proteins, lipids, and nucleic acids. MG-induced cellular stress has been comprehensively studied in the pathogenesis of diabetes, where MG-derived advanced glycation end products (MG-AGES) notably contribute to diabetes initiation and the development of microvascular complications [16]. Our team and others have demonstrated that MG-AGES, hydroimidazolones (MGHs), and argpyrimidine (Argp) adducts are a common feature of human breast and colon tumors, with increased expression in tumors compared to normal tissue [17,18,19]. We have shown that breast cancer cells stably silenced for glyoxalase 1 (GLO1), the main MG detoxifying enzyme, showed enhanced tumor growth and metastasis development, which were facilitated by heat shock protein 90 (HSP90) glycation and inhibition of the Hippo tumor suppressor pathway [20]. Furthermore, upon GLO1 depletion, breast cancer cells displayed a distinctive MG stress gene signature, which highlighted major ECM remodeling and activation of migratory and invasion signaling pathways [21]. In colon cancer, MG stress enhanced tumor growth in vivo, an effect that was efficiently blocked using carnosine (beta alanyl-L-histidine), a potent scavenger of MG [17]. Other reported MG scavengers include the anti-hyperglycemic drug metformin [22,23] and aminoguanidine [24]. KRAS-mutated colorectal tumor cells are generally glycolytic and intrinsically resistant to EGFR-targeted therapy, such as cetuximab. We have previously shown that these cells were efficiently resensitized to cetuximab, in vitro and in vivo, through inhibition of MG stress using carnosine [25]. Altogether, these studies demonstrated that MG, kept under a sub-cytotoxic level [26], not only promotes cancer but interferes with response to anti-cancer therapies. In support of its pro-oncogenic functions, MG joined the critical group of metabolic intermediates, known as oncometabolites, whose altered levels play a major role in cancer [27,28,29]. The extent to which oncometabolites interfere with cancer therapy, and how they might be exploited to improve patients’ response to treatment, is currently the object of intense investigation.

The heat shock response (HSR) is an adaptive cellular process that functions to maintain the proteome during elevated temperatures and other forms of proteotoxic stresses. Under these latter conditions, heat shock proteins (HSPs) play the role of molecular chaperones, maintaining proteins in their folded functional state. The rapid expression of key HSPs, e.g., HSP90, HSP70, and HSP27, are triggered by heat shock factor 1 (HSF1), a highly conserved transcription factor that mediates HSR. Next to classical HSR, HSF1 is also involved in oncogenic transformation, proliferation, metastatic dissemination, and anti-cancer drug resistance (for review, [30]).

MG-mediated glycation not only contributes to the accumulation of modified proteins but also affects their proteasomal degradation, both directly and indirectly. Bento et al. have shown that MG impairs proteasome activity and leads to accumulation of ubiquitin conjugates in human retinal pigment epithelial cells [31]. They also noted elevated transcriptional activity of HSF1, indicating that MG-stressed cells promote HSF1 activity to cope with protein glycation stress. Additionally, HSP27 is a target of MG in cancer cells and its anti-apoptotic function has been well established in the context of MG-mediated glycation stress. Indeed, glycated HSP27 proved to be more efficient than native HSP27 in assisting the escape of tumor cells from apoptosis [32,33]. A recent study using pancreatic cancer cells reported that HSF1 inhibition sensitized cells to gemcitabine [34].

In this study, we establish a fundamental link between gemcitabine resistance, HSR and MG stress occurrence in glycolytic PDAC tumors. We found that MG stress is more detectable in gemcitabine-treated pancreatic tumors compared to untreated ones, and is associated with poor patient outcome. In PDAC models of acquired gemcitabine resistance, MG favors cell survival and gemcitabine tolerance, through the regulation of HSR. This novel connection lets us propose the use of MG scavengers, such as metformin and aminoguanidine, to prevent the unintended consequences of glycolytic rewiring, i.e., MG stress and subsequent activation of HSR, in tumors of PDAC patients receiving gemcitabine therapy.

## 2. Materials and Methods

### 2.1. Cell Lines, Culturing, and Chemicals

This study utilized human pancreatic cancer cells (MiaPaca2, Patu-8988T, and T3M4) and human pancreatic normal epithelial cells (HPNE). The MiaPaca2 and HPNE cell lines were purchased from ATCC (Manassas, VA, USA), the Patu-8988T cell line was purchased from DSMZ (Braunschweig, Germany), and the T3M4 cell line was a generous gift from Professor Pankaj Singh (University of Nebraska Medical Center, Omaha, NE, USA). All human pancreatic cancer cells were cultured in high glucose Dulbecco’s Modified Eagle Medium (DMEM). MiaPaca2 and T3M4 cells were supplemented with 10% FBS, 4 mM L-glutamine, and 1 mM sodium pyruvate. Patu-8988T cells were supplemented with 5% FBS, 5% horse serum, and 2 mM L-glutamine. HPNE were cultured according to manufacturer’s recommendations. All cells were incubated at 37 °C in a humidified chamber with 5% CO_2_. All cell lines were authenticated using STR profiling at Leibniz-Institute DSMZ (Braunschweig, Germany) and were regularly checked for mycoplasma contamination using MycoAlert mycoplasma detection kit (Lonza, Basel, Switzerland). Short-term gemcitabine treatments consisted of low-dose gemcitabine treatments applied 2 times a week, for 0 to 2 months, to parental PDAC cells (MiaPac2, Patu-8988T, and T3M4), at steadily increasing doses. After 1 month, parental PDAC cells were able to proliferate in 20–50 nM of gemcitabine, and this was increased to 80–100 nM of gemcitabine after 2 months. Gemcitabine-resistant (GR) PDAC cell lines (MiaPaca2 and T3M4) were generated from parental cells by steadily increasing the dose of gemcitabine in culture over 10 to 12 months. Surviving cells were allowed to recover between gemcitabine challenges. Gemcitabine resistance was determined as the concentration of gemcitabine in which treated cells showed approximately 10-fold higher proliferation compared to parental cells (as calculated by inhibitory concentration 50 (IC50) analysis). GR-MiaPaca2 and GR-T3M4 cells reached the desired resistance to gemcitabine at 450 nM and 900 nM, respectively. Gemcitabine (Cat# G-6423), methylglyoxal (Cat# M-0252), aminoguanidine (Cat# 396494), and metformin (Cat# D-150959) were all obtained from Sigma-Aldrich (Bornem, Belgium). The antibodies used for Western blotting and immunohistochemistry (IHC) are listed in Appendix A.

### 2.2. Clinical Tumor Samples

Retrospective series of pancreatic ductal adenocarcinoma (PDAC) primary tumors were provided by Prof Van Laethem (Free University of Brussels, Brussels, Belgium). Ethical approval for the use of the tumors for this research has been approved by Committee of Medical Ethics—Erasme Hospital (approval number P2021-356). The cohort of 12 PDAC cancer patients consisted of 6 gemcitabine-treated (neo-adjuvant chemotherapy) and 6 non-treated tumors. The median age at diagnosis was 70.95 years (range: 50–87.7 years). Sex, age, pTMN staging, and survival data were retrieved from medical reports (Table 1). For TNM prognostic stage determination, we referred to the 7th edition of the American Joint Committee on Cancer and Union Internationale Contre le Cancer (AJCC-UICC).

### 2.3. Immunohistochemistry (IHC)

Formalin-fixed paraffin-embedded sections were deparaffinized in xylene and rehydrated. Endogenous peroxidase activity was blocked by treating sections with 3% hydrogen peroxide in methanol for 30 min and subsequent washing in PBS for 20 min. Sections were incubated in sodium citrate buffer (10 mM; pH 6) for 40 min at 95 °C for antigen retrieval and 1.5% normal serum (Vector Laboratories, Newark, CA, USA) for 30 min to block non-specific serum-binding sites. Next, sections were incubated with mouse anti-MGHs (Cell Biolabs, 1:1000), mouse anti-GLO1 (BioMAC, 1:500 dilution) and rabbit anti-HSP27 (Enzo Life Sciences, 1:2000 dilution) antibodies overnight at 4 °C. Antibody binding was detected using anti-mouse or anti-rabbit biotinylated secondary antibody (Vector Laboratories) incubated on sections for 30 min at room temperature. Next, sections were incubated with the avidin-biotin-peroxidase complex (Vectastain ABC Kit; Vector Laboratories) for 30 min followed by immunoreactivity revelation with 3,30 diaminobenzidine tetrachloride. Finally, sections were counterstained with hematoxylin, dehydrated, and mounted with DPX (Sigma-Aldrich, Bornem, Belgium). Negative control tissue sections incubated without primary antibody showed no detectable immunoreactivity. The immunostaining was reviewed and scored by two examiners including an anatomical pathologist (N.D). All slides of human PDAC samples were stained with hematoxylin and eosin and reviewed by the same experienced pathologist (N.D) to assess for the presence of tumoral areas adequate for IHC. Scoring of the staining was calculated according to the intensity of the staining (none (0), weak (1+), moderate (2+), high (3+)). For tumor cases presenting with heterogeneity in staining intensity, the scoring was performed according to the staining of the most positive tumor cells, when their estimated percentage represented at least 30% of the total positive tumor cell area evaluated. For MGHs, GLO1, and HSP27 staining, scores of 0 to 2 were considered as low to intermediate staining (low/intermediate MG stress or heat shock response) and scores of 3 were considered as high staining (high MG stress or heat shock response).

### 2.4. Gemcitabine Inhibitory Concentration 50 (IC50) Determination

Cells were plated in 24-well plates and treated for 72 h with increasing doses of gemcitabine. Cells were then washed, lysed by sonication, and their DNA content was assessed using bisbenzimide (Sigma) incorporation. DNA content was detected spectrophotometrically by excitation at 360 nm and fluorescence emission at 460 nm. IC50 was determined as the concentration of gemcitabine able to decrease the quantity of DNA detected by half.

### 2.5. Extracellular Flux Analysis

MiaPaca2 (10,000 cells/well) and T3M4 (10,000 cells/well) parental cells were seeded in Seahorse XFp mini-plates (Agilent, Santa Clara, CA, USA) and analyzed using the Seahorse glycolysis stress test according to manufacturer’s recommendations. Cells were first challenged with glucose (10 mM) then successively stressed with oligomycin (1 μM) and 2-deoxyglucose (50 mM). All results were normalized to cell number using bisbenzimide (Sigma) incorporation. Extracellular flux analyses were performed on three independent biological replicates.

### 2.6. Western Blotting

Cell protein was extracted in 1% SDS buffer containing protease and phosphatase inhibitors (Roche, Basel, Switzerland). Protein concentrations were determined using the bicinchoninic acid (BCA) assay (Pierce, Waltham, MA, USA). Twenty µg of proteins was separated using 7.5 to 12.5% sodium dodecyl sulfate polyacrylamide gel electrophoresis (SDS-PAGE) and transferred to a methanol-activated PVDF membrane. After transfer, membranes were blocked in TBS-Tween 0.1% containing 5% non-fat dried milk (Bio-Rad, Hercules, CA, USA) and were incubated overnight at 4°C with the primary antibodies listed in Appendix A. Anti-argpyrimidine antibody (mAb6B) specificity has been previously confirmed by competitive enzyme-linked immunosorbent assay (ELISA) and shown to not react with other MG-arginine adducts, such as 5-hydro-5-methylimidazolone and tetrahydropyrimidine (Oya et al., 1999). Subsequently, membranes were incubated with horseradish peroxidase-conjugated secondary antibodies (anti-rabbit or anti-mouse) for 1 h at room temperature. The immunoreactive bands were visualized using Enhanced Chemiluminescence Western Blotting substrate (Pierce), were quantified with densitometric analysis, and normalized for β-actin using ImageJ software Version 1.53. Protein expression data are presented as mean ± SEM of three independent biological replicates.

### 2.7. Cell Proliferation Assay

Real-time proliferation was assessed in metformin-, aminoguanidine- or gemcitabine-treated MiaPaca2 and T3M4 (parental compared to gemcitabine-resistant (GR)) cell lines, using the IncuCyte S3 Live-Cell Analysis System. Cells were seeded in triplicate into 96-well plates (3000 cells/well). After an overnight incubation, cells were treated with either metformin (10 mM), aminoguanidine (10 mM), or gemcitabine (50 nM), and the plate was inserted into the IncuCyte S3 Live-Cell Analysis System for real-time imaging. Phase-contrast images were automatically acquired using a 10× objective lens (four images per well) at 24 h intervals over 72 h. The IncuCyte Analyzer provides real-time cellular confluence data based on segmentation of high-definition phase–contrast images. Cell proliferation is expressed as an increase in confluency (percentage) relative to the cell confluence quantified at the start of the experiment (0 h). IncuCyte assays were performed on two independent biological replicates.

Cell proliferation in metformin-treated T3M4 and GR-T3M4 cells was assessed using bisbenzimide (Sigma) incorporation. Cells were plated in 24-well plates (30,000 cells/well) and treated for 24 h with increasing doses of metformin (0, 1, 5, and 10 mM). Cells were then washed, lysed by sonication, and their DNA content was assessed using bisbenzimide incorporation. DNA content was detected spectrophotometrically by excitation at 360 nm and fluorescence emission at 460 nm. Proliferation was determined as the quantity of DNA relative to untreated parental or GR cells.

### 2.8. GLO1 Activity

GLO1 activity was assessed in MiaPaca2 and T3M4 (parental compared to gemcitabine-resistant (GR)) cell lines, as previously described [25]. Briefly, proteins were extracted from cells with radioimmunoprecipitation (RIPA) buffer, quantified by BCA assay (Pierce), and mixed with a pre-incubated (15 min at 25 °C) equimolar (1 mM) mixture of MG and GSH (Sigma) in 50 mM sodium phosphate buffer (pH 6.8). S-D-lactoylglutathione formation was followed spectrophotometrically by the increase in absorbance at 240 nm. GLO1 activity data are expressed as arbitrary units (A.U.) of enzyme per milligram of proteins. GLO1 activity data are presented as mean ± SEM of three independent biological replicates.

### 2.9. L-Lactate Production

L-lactate concentrations were assessed in conditioned medium from MiaPaca2 and T3M4 (parental compared to gemcitabine-resistant (GR)) cell lines. Conditioned media were diluted 3 times and incubated in the presence of NAD+, hydrazine and L-lactate dehydrogenase enzyme (Sigma) for 30 min. L-lactate concentration was determined by comparing NADH formation, measured spectrophotometrically by an increase in absorbance at 320 nm, to a L-lactate calibration curve. L-lactate concentrations were normalized per million cells, and fold changes were calculated by comparing to parental cells.

### 2.10. RNA Isolation and Quantitative Reverse Transcription PCR (qRT-PCR)

RNA extraction was performed using the NucleoSpin RNA extraction kit (Macherey-Nagel, Düren, Germany) according to manufacturer’s instructions. cDNA was generated by reverse transcription using the Transcription First Strand cDNA Synthesis Kit (Roche). qPCR reactions were set up with 400 ng of cDNA, 300 ng of each primer (HSPB1 5′-CTGACGGTCAAGACCAAGGATG-3′ (forward) and 5′-GTGTATTTCCGCGTGAAGCACC-3′ (reverse); HSP90AA1 5′-GTCCTGTGCGGTCACTTAGC-3′ (forward), and 5′-AAAGGCGAACGTCTCAACC-3′ (reverse)), and 1X Fast Start SYBR Green Master Mix (Roche). qPCR was performed using the 7300 Real Time PCR System and corresponding manufacturer’s software (Applied Biosystems, Carlsbad, CA, USA). Relative gene expression was normalized to 18S rRNA. Primers were synthesized at Eurogentec (Seraing, Belgium).

### 2.11. Oncogenic Mutation Analysis

The regions of interest were amplified using the Tumor Hotspot MASTR Plus kit (Agilent, USA), according to the manufacturer’s protocol. Sequencing was performed on a MiSeq platform using v3 chemistry with 2 × 250 bp cycles (Illumina, San Diego, CA, USA). Sequencing files were analyzed with the MASTR Reporter software (Agilent, Santa Clara, CA, USA).

### 2.12. In Silico Analysis

The Cancer Genome Atlas (TCGA) PanCancer data retrieved from cBioPortal (http://www.cbioportal.org/ accessed on 13 December 2022), was utilized to assess gene expression of important glycolytic enzymes (GLUT1, HK1, G6PD, ALDOA, TPI1, GAPDH, PGK1, ENO1, PKM, and LDHA), glyoxalase 1 (GLO1), and heat shock response-associated proteins (HSP27, HSP90, and HSF1), in order to analyze potential associations of tumor MG stress levels with the heat shock response and patient survival across *n* = 179 PDAC tumors. The expression (microarray z-scores) of glycolytic enzymes and GLO1 were used to split PDAC tumors into low (*n* = 59) or high (*n* = 57) MG stress groups. The difference in survival (OS and DFS) and heat shock response were compared between the PDAC tumors with either high or low MG stress.

### 2.13. Statistical Analysis

Mean and standard error of the mean (SEM) were calculated for data from independent replicate experiments. Normal distribution of data was assumed for cell culture studies, as normality testing is not able to be performed on replicate numbers of *n* = 3. Testing for normal distribution of patient tumor data (TCGA) was performed using KS normality testing. Statistical significance between groups or experimental conditions were tested by paired and unpaired two-tailed T-testing (Mann–Whitney U for non-normal data), One-way ANOVA with Tukey’s Multiple Comparison Testing (Wilcoxon Rank Sum test for non-normal data), and Two-way ANOVA with post hoc Bonferroni correction. Correlational data were evaluated using Spearman’s correlation coefficient. IC50 concentrations of chemotherapeutic agents were determined from non-linear dose–response curves that were fitted using the four-parameter variable slope model. Statistical significance was considered as a *p*-value < 0.05. All data analysis was performed in GraphPad Prism Version 5.01.

## 3. Results

### 3.1. MG Stress Is Associated with Poor Patient Outcome in PDAC

To evaluate the influence of MG stress on pancreatic cancer patient outcomes, we analyzed the association between expression of MG stress markers and patient survival across PDAC tumors (*n* = 179) using The Cancer Genome Atlas (TCGA) database. Markers of MG stress included both the glycolytic capacity of the tumors, as MG is spontaneously generated during glycolysis, and the expression of the key glyoxalase system enzyme GLO1, responsible for facilitating the detoxification of MG (Figure 1A). Firstly, according to the median gene expression, each patient was classified as high- or low-expressing for 10 important glycolytic enzymes/transporters (GLUT1, HK1, GPI, ALDOA, TPI1, GAPDH, PGK1, ENO1, PKM, and LDHA) (Figure 1B) (Appendix A). Next, an overall glycolysis score was calculated for each tumor based on the number of times that tumor sample appeared in the high expression group (above median expression) for each glycolysis gene (Figure 1C). PDAC tumors with glycolysis scores of 8–10 or 0–2 were established as either the high (*n* = 57) or low (*n* = 59) glycolytic tumors, respectively (Figure 1C). Further, we combined glycolysis scores with GLO1 expression to identify high and low MG-stress tumors (Figure 1D). Interestingly, PDAC tumors with high MG stress showed significantly shorter overall and disease-free survival compared to low MG-stress tumors (Figure 1E,F), suggesting that MG stress promotes disease progression, potentially by interfering with the efficacy of therapy.

### 3.2. Gemcitabine Increases MG Stress in Gemcitabine-Sensitive Pancreatic Cancer Cells

Next, we wanted to evaluate whether pancreatic cancer cells, with differing gemcitabine sensitivities and glycolytic capacities, exhibit MG stress in response to a gemcitabine challenge. Firstly, we used three KRAS mutant pancreatic cancer cell lines (Patu-8988T, MiaPaca2 and T3M4) and one normal human epithelial pancreatic cell line (HPNE), and compared basal gemcitabine sensitivity using IC50 analysis, assessing cell proliferation after a 72 h treatment. Of the pancreatic cancer cell lines under study, MiaPaca2 showed the highest level of tolerance to gemcitabine (IC50 = 21.8 nM [95% CI 18.45–25.74]), whereas Patu-8988T (IC50 = 6.3 nM [95% CI 5.72–6.96]) and T3M4 (IC50 = 5.4 nM [95% CI 4.29–6.84]) exhibited a similar sensitivity to gemcitabine as HPNE cells (IC50 = 4.1 nM [95% CI 3.45–4.96]); *p* < 0.0001 for sum of squares F-test comparing best fit values of MiaPaca2 to Patu-8988T, T3M4, and HPNE (Figure 2A). Subsequently, we examined whether baseline sensitivity to gemcitabine was associated with rates of glycolysis, and potentially MG production, by performing a quantitative measurement of the glycolytic flux of MiaPaca2, T3M4, and Patu-8988T cells. Gemcitabine-tolerant MiaPaca2 cells presented with significantly higher glycolytic capacity upon glucose challenge than gemcitabine-sensitive T3M4 and Patu-8988T cells (Figure 2B). Having shown that tolerance to gemcitabine and glycolysis are lower in T3M4 and Patu-8988T compared to MiaPaca2 cells, we examined whether a short-term gemcitabine challenge (weekly exposures to represent clinical practice) effects MG stress levels differently between cell lines (as depicted in Figure 2C). Western blot detection of argpyrimidine MG adducts (Argp) after a gemcitabine challenge showed a significant time-dependent accumulation in T3M4 and Patu-8988T cells, but not in MiaPaca2 cells (Figure 2D), when compared to untreated parental cells. However, the increase in Argp expression induced by short-term gemcitabine exposure was greater in T3M4 than Patu-8988T cells (Figure 2D). Similar to Argp, expression of the MG detoxification enzyme GLO1 was significantly increased in T3M4, but not Patu-8988T and MiaPaca2 cells, after 1 and 2 months of gemcitabine challenge (Figure 2E). These results indicate that gemcitabine-tolerant MiaPaca2 cells bear enhanced glycolytic capacity when compared to gemcitabine-sensitive T3M4, Patu-8998T, and normal cells. Additionally, in response to a clinically relevant gemcitabine challenge, initially sensitive and low-glycolytic pancreatic cancer cells undergo an early and significant increase in protein glycation.

### 3.3. MG Stress Is Associated with Gemcitabine Resistance in PDAC

Our data suggest that, in PDAC cells, gemcitabine response is related to rates of glycolysis and MG stress. Therefore, we next aimed to assess whether glycolysis and MG stress undergo significant changes upon acquired gemcitabine resistance in PDAC cells presenting with differing sensitivities to gemcitabine and rates of glycolysis at baseline. We began by rendering gemcitabine-sensitive and tolerant PDAC cell lines, T3M4 and MiaPaca2, respectively, resistant to gemcitabine by chronic and repeated exposure to increasing gemcitabine concentrations (Figure 3A). Gemcitabine-resistant (GR) cells were validated using IC50 analysis, comparing parental and GR cell proliferation after a 72 h gemcitabine treatment. Compared to parental cells, GR-MiaPaca2 and GR-T3M4 cells showed greater than 10-fold higher resistance to gemcitabine (Figure 3B). Using next-generation sequencing of genomic DNA from parental and GR cells, we assessed the status of KRAS, and other key oncogenes, during gemcitabine resistance acquisition (Appendix A). Approximately 90% of pancreatic tumors present with mutated KRAS [12]. Mutant KRAS supports proliferation and therapy resistance [35] and is typically associated with glycolytic rewiring in PDAC [13]. MiaPaca2 and T3M4 cells harbor clinically relevant KRAS G12C and Q61H mutations, respectively. Importantly, KRAS G12C and Q61H mutations were retained during acquisition of gemcitabine resistance in MiaPaca2 and T3M4 cells, respectively, while no other major oncogenic activating mutations were evidenced (Appendix A).

Next, parental and GR cell pairs were utilized to characterize the effect of acquired gemcitabine resistance on the glycolytic phenotype of the cells, as MG is spontaneously generated during glycolysis [16] and enhanced glycolysis has been associated with gemcitabine resistance in PDAC [6,10,11]. Western blot analysis confirmed an upregulation of glucose transporter 1 (GLUT1) and lactate dehydrogenase (LDHA) in GR-T3M4, but not GR-MiaPaca2 cells (Figure 3C). Moreover, L-lactate (the final product of glycolysis) accumulation was elevated in GR-T3M4 compared to parental T3M4 cells, but GR-MiaPaca2 had a similar L-lactate level to parental MiaPaca2 cells (Figure 3D). These findings suggest that enhanced glycolytic activity in GR-T3M4 is most likely accompanied by increased MG stress. In good accordance with this hypothesis, GR-T3M4 cells had elevated accumulation of Argp adducts when compared to their corresponding parental line (Figure 3E).

Consistent with their greater glycolytic potential, basal levels of MG adducts were elevated in MiaPaca2 compared to T3M4 parental cells (Figure 3E). In line with these findings, GR-T3M4, but not GR-MiaPaca2 cells, showed elevated levels of GLO1 expression (Figure 3F) and activity (Figure 3G) relative to their parental lines. Increased rates of GLO1 detoxification capacity validates the presence of elevated MG stress in the GR-T3M4 cells. Finally, to determine whether GR-T3M4 cells depend on elevated MG stress for proliferation, we treated T3M4 cells (parental and GR) with increasing doses of metformin, a potent MG scavenger, and assessed cell proliferation rates with Hoechst incorporation. After 24 h, metformin induced a dose-dependent inhibitory effect on GR-T3M4 cell proliferation that was significantly more important in GR-T3M4 compared to parental T3M4 cells (Figure 3H), suggesting that MG-stressed GR-T3M4 cells rely more on MG for their proliferation than parental cells. Taken together, these and previous data suggest that T3M4 cells undergo an increase in MG stress, likely through elevated glycolysis, while acquiring resistance to gemcitabine. While the higher basal levels of glycolysis and MG stress present in MiaPaca2 cells (when compared to T3M4) likely provide an environment for lowered gemcitabine sensitivity, the acquisition of chronic gemcitabine resistance in GR-MiaPaca2 is possibly driven through MG-independent mechanisms.

### 3.4. MG Stress Mediates Gemcitabine Resistance Acquisition through Regulation of the Heat Shock Response

Having shown that MG stress is associated with gemcitabine challenge and acquired resistance in initially sensitive PDAC cells, our next aim was to investigate potential molecular mechanisms by which MG stress promotes acquisition of gemcitabine resistance. Interestingly, accumulation of MG adducts in GR-T3M4 cells (Figure 3E) was predominantly represented by glycation of protein (s) with an approximate molecular weight of 25 kDa, most likely corresponding to the well-described MG target protein, heat shock protein 27 (HSP27). We [20,25], and others [32,33,36], have previously demonstrated that the heat shock proteins HSP27 and HSP90 are detectable as argpyrimidine MG adducts in other cancer types. In this study, we have shown that basal expression of HSP27 and HSP90 proteins (Figure 4A) and mRNA (HSPB1 and HSP90AA1, respectively) (Appendix A) are higher in MiaPaca2 compared to T3M4 parental cells, correlating with higher gemcitabine resistance of MiaPaca2 cells. Additionally, HSP27 and HSP90 expression increased significantly in GR-T3M4, but not in GR-MiaPaca2, compared to parental cells, during acquisition of gemcitabine resistance (Figure 4A,B). We next showed that gene expression of HSPB1 and HSP90AA1 were indeed upregulated by acute MG treatment in parental T3M4 but not MiaPaca2 cells, while gemcitabine treatment could not match this effect (Figure 4C,D). These observations indicate that increased HSP27 and HSP90 expression in GR-T3M4 cells (Figure 4A,B) are likely a consequence of chronic gemcitabine pressure triggering both glycolysis and MG stress.

Importantly, although MG glycation of HSP27 has previously been shown to contribute to oligomerization and anti-apoptotic activity of HSP27 [32], the role of MG in HSP27 (HSPB1) gene expression regulation is novel to this study. Using TCGA data, we highlighted the variable gene expression but low methylation levels of HSP27 across 179 PDAC tumors (Appendix A), suggesting that HSP27 expression in PDAC is likely regulated by methylation-independent mechanisms. Moreover, the strong positive correlation observed between HSP27 and heat shock factor 1 (HSF1) gene expression, the well-established and predominant transcription factor regulating heat shock proteins (including HSP27 and HSP90) (Appendix A), highlights a potential mechanism by which MG may be regulating heat shock protein expression. Accordingly, HSF1 abundance was elevated in MiaPaca2 compared to T3M4 parental cells, and was significantly increased in GR-T3M4, but not GR-MiaPaca2, cells (Figure 4E); this correlates with data on MG stress and heat shock protein expression. These results highlight a potentially novel mechanism of MG-mediated heat shock response (HSR) regulation, through induction of HSF-1, which may contribute to gemcitabine resistance acquisition in PDAC.

In order to further support these original findings and assess their clinical relevance, we first checked whether MG stress is associated with HSR using publicly available PDAC data (TCGA), and second whether gemcitabine exposure influences MG stress and HSR abundance in our own cohort of PDAC tumors (outlined in Table 1). We utilized the high- and low-MG-stress tumors (outlined in Figure 1D; TCGA data), to elucidate for the first time a significant association between high MG stress and increased mRNA expression of HSF1, HSP27, and HSP90 in PDAC (Figure 5A). Furthermore, we performed immunohistochemical analysis of MG stress (GLO1 and MGHs MG adducts) and HSR (HSP27) markers on our cohort of pancreatic cancer tumors treated with (*n* = 6) or without gemcitabine (*n* = 6) prior to tumor resection (Table 1). MGHs and HSP27 were detected in the cytoplasm of PDAC cells, while GLO1 showed a mixed cytoplasmic and nuclear staining pattern. The intensity of the cytoplasmic immunostaining was evaluated for the scoring. We showed a trend towards increased MG stress and HSR in gemcitabine-treated compared to untreated PDAC tumors (Figure 5B,C). Taken together, these results suggest that, during gemcitabine treatment, increased MG stress may be a mechanism by which PDAC tumors build resistance to gemcitabine by upregulating the HSR, leading to cell survival.

### 3.5. MG Scavenging Inhibits Gemcitabine-Resistant PDAC Cell Proliferation and Expression of Heat Shock Response Proteins

In order to explore whether MG stress could be a targetable vulnerability of gemcitabine-resistant PDAC, we treated GR cells (T3M4 and MiaPaca2) with potent MG scavengers, metformin and aminoguanidine, and assessed cell proliferation over time. Compared to untreated cells, metformin and aminoguanidine significantly inhibited proliferation of GR-T3M4 cells, whereas only metformin, but not aminoguanidine, was able to significantly inhibit proliferation of GR-MiaPaca2 cells after 72 h (Figure 6A). Moreover, the greater impact of MG scavengers on GR-T3M4 cell proliferation, as compared to GR-MiaPaca2, highlights the fact the MG stress is driving resistance acquisition in GR-T3M4 cells, but not in GR-MiaPaca2 cells, supporting our earlier findings. To validate that the HSR is the molecular mechanism by which MG promotes viability in gemcitabine-resistant PDAC cells, we compared expression of heat shock-related proteins (HSF1, HSP27, and HSP90) in MG-stressed GR-T3M4 cells, treated with or without MG scavengers (metformin and aminoguanidine). After 24 h, aminoguanidine (30 mM) significantly reduced expression of HSF1, HSP27, and HSP90, and metformin (30 mM) significantly decreased expression of HSF1 and HSP90, in GR-T3M4 cells (Figure 6B). Although not reaching significance, HSP27 expression was also reduced by metformin (5 and 30 mM) (Figure 6B). Moreover, we observed a significant difference in expression of HSF1, HSP27, and HSP90 between cells treated with 5 or 30 mM aminoguanidine (Figure 6B), suggesting the effect of MG scavenging on HSR expression is dose dependent and mirrors the effect of MG scavengers on GR-T3M4 cell proliferation (Figure 6A). Taken together, these results suggest that MG scavengers could be used to inhibit the MG/HSR axis to tackle gemcitabine resistance in PDAC patients.

## 4. Discussion

The molecular mechanisms by which gemcitabine challenge ultimately induces acquired resistance remain under debate. Thus, the development of strategies to improve gemcitabine efficacy in PDAC treatment is limited. In order to satisfy the demand of growth and proliferation, glycolytic rewiring, also known as the Warburg effect, is triggered upon oncogenic alterations in PDAC tumors. Almost ten years ago, aberrant activation of the KRAS pathway was shown to promote glucose avidity in pancreatic tumors, whereby KRAS mutations were associated with the upregulation of the glucose transporter GLUT1, and enzymes that execute aerobic glycolysis, such as LDHA [13,37]. More recently, increased glycolytic flux was associated with tolerance to gemcitabine through a novel mechanism, in which de novo pyrimidine biosynthesis is in molecular competition with gemcitabine, and reduces its efficacy in T3M4 and MiaPaca2 PDAC models [6]. Interestingly, the same study pointed to an increase in the intermediate metabolites from the upper reactions of the glycolytic pathway. Specifically, they observed elevated levels of the metabolites upstream of dihydroxyacetone phosphate and glyceraldehyde-3-phosphate in gemcitabine-resistant cells [6]. Knowing that MG is formed from the spontaneous dephosphorylation of these specific triose phosphate intermediates, this observation is in good accordance with our data, showing that the production of MG is favored in PDAC after gemcitabine challenge.

In this study, we first validated the association between the gemcitabine-resistant phenotype and glucose metabolism with the occurrence of MG stress in three PDAC cell line models. We have previously demonstrated that KRAS mutant glycolytic cancer cells challenged with 13C-glucose produce labeled MG and accumulate MG protein adducts [25]. Here, we observed that a 1 month challenge with gemcitabine already induced MG adduct accumulation, which was paralleled by an adaptation of the cells to MG stress through elevated GLO1 expression. Interestingly, MiaPaca2 cells, characterized here as more glycolytic and tolerant to gemcitabine at baseline, when compared with T3M4 and Patu-8988T cells, did not show increased MG adduct formation or GLO1 adaptation. Thus, it is likely that MiaPaca2 cells have previously undergone these changes or their resistance was dependent on mechanisms other than MG stress induction. In good accordance with these hypotheses, the corresponding long-term challenge clones (described herein as ‘GR’ clones) showed a glycolytic rewiring, assessed through GLUT1 and LDHA induction, that occurred primarily in sensitive T3M4 cells and not in the more gemcitabine-tolerant MiaPaca2 cells. It is noteworthy that GR-T3M4 showed elevated MG adducts and GLO1 expression that were comparable to the basal levels observed in MiaPaca2 parental cells.

Next, we reasoned that cancer cells with high rates of glycolysis produce MG and its protein adducts, and thus require a relatively high rate of MG detoxification for survival. Accordingly, GLO1 is upregulated in the more glycolytic cancerous pancreatic tissues, than the related non-cancerous tissues [38]. Adding further clinical relevance, we observed that PDAC patients presenting with tumors that have elevated glycolytic scores and GLO1 levels show poorer overall and disease-free survival rates. This extends our previous observation of MG stress being a pejorative feature associated with cancer progression and metastasis in other cancer types, e.g., breast and colon adenocarcinoma.

Mechanistically, we show that MG-mediated gemcitabine resistance occurs through the activation of HSR in PDAC cells. This novel finding in the context of tolerance to gemcitabine therapy is well supported by several previous studies that have contributed to the establishment of a link between MG stress and key heat shock stress effectors, such as HSP27 and HSP90. In fact, HSPs are the best-studied targets of MG-mediated glycation in PDAC, as well as other cancer types [25,32,33,36]. We have shown that HSP90 is glycated on several arginine residues in stably GLO1-depleted breast cancer cells, which impeded the good functioning of the Hippo tumor suppressor pathway [20].

In addition to its classical functions during HSR, HSP27 has aroused great interest in the context of cancer, and more recently in gemcitabine resistance studies. Several studies have reported increased HSP27 in various types of malignant tumors, where HSP27 displayed cancer-promoting functions and was associated with poor prognosis in cancer patients [39]. In fact, elevated HSP27 favors cancer cell growth and metastasis, inhibition of apoptosis, and chemoresistance [40]. A proteomic study conducted on gemcitabine-resistant and -sensitive human PDAC cell lines, pointed to HSP27 as a key protein associated with resistance to gemcitabine [41]. Interestingly, the silencing of HSP27 in MiaPaca2 PDAC cells inhibits proliferation, induces apoptosis, and enhances chemosensitivity to gemcitabine, thus indicating its significant implication in the tolerance to this drug [42]. Consistently, elevated HSP27 in tumor specimens was associated with higher resistance to gemcitabine and shorter survival in PDAC patients [41,43]. As such, HSP27 is considered as a critical therapeutic target for effective cancer therapy [44,45].

Based on this cumulative evidence, we consider that the control of HSP27 levels and activity, via MG stress, is of the utmost importance. In this study, we observed that HSP27 was significantly induced upon acquired gemcitabine resistance and that it was detectable under its glycated form using specific anti-Argp antibodies. Based on these data, we propose that the expression level of key HSPs is tightly controlled under MG stress, both at the mRNA and protein levels, in PDAC cells following gemcitabine challenge (Figure 6C). In fact, we showed that gemcitabine-resistant cells present with increased HSF1 transcription factor levels, which revert in the presence of MG scavengers. It is likely that elevated protein levels of HSP27 result both from augmented transcription (this study) and through protein stabilization mechanisms, leading to an increased half-life of MG-glycated HSP27 protein, as we have previously shown in colon cancer [25]. Consistently, PDAC tumors with high MG stress scores demonstrated significantly elevated HSP27 and HSP90 when compared with low-MG-stress tumors. More specifically, we have shown a trend toward the detection of higher levels of HSP27, MG adducts, and GLO1 in gemcitabine-treated versus untreated PDAC tumor samples using immunohistochemistry. No doubt, a larger series of tumor samples will be necessary in the future to confirm MG stress-induced HSR as an adverse effect of gemcitabine treatment in PDAC.

Importantly, we further evidenced the dependence of gemcitabine-resistant cells on MG stress using MG scavengers. We showed that resistant cell growth is significantly affected under metformin and aminoguanidine treatment. Under this condition, we also observed a reduction in HSR, represented by a significant decrease in HSF1, HSP27, and HSP90 protein expression. HSP27 response is not specific to gemcitabine challenge, as it has been also implicated in resistance to other drugs, such as doxorubicin [46]. Thus, the HSR activation, notably resulting in the anti-apoptotic functions of HSP27, could rather be considered as a global response to chemical stress, such as therapy. Future studies will help to position MG stress and understand its connections with other cellular conditions (i.e., oxidative stress and proteotoxic stress) that favor cancer cell survival, by means of certain adaptive changes arising under the pressure of anti-cancer therapy.

To conclude, it is worth highlighting that MG production occurs unavoidably when glycolysis is increased, and this by-product of glycolysis must not be underestimated. MG should be considered as an important trigger of major cellular stress contributing to cancer progression and therapy resistance. This study paves the way for future pre-clinical and clinical studies utilizing already available and efficient MG scavengers, such as metformin and aminoguanidine, either alone or in combination, to overcome gemcitabine resistance in pancreatic cancer through the MG blockade.

## Figures and Tables

**Figure 1 cells-12-01414-f001:**
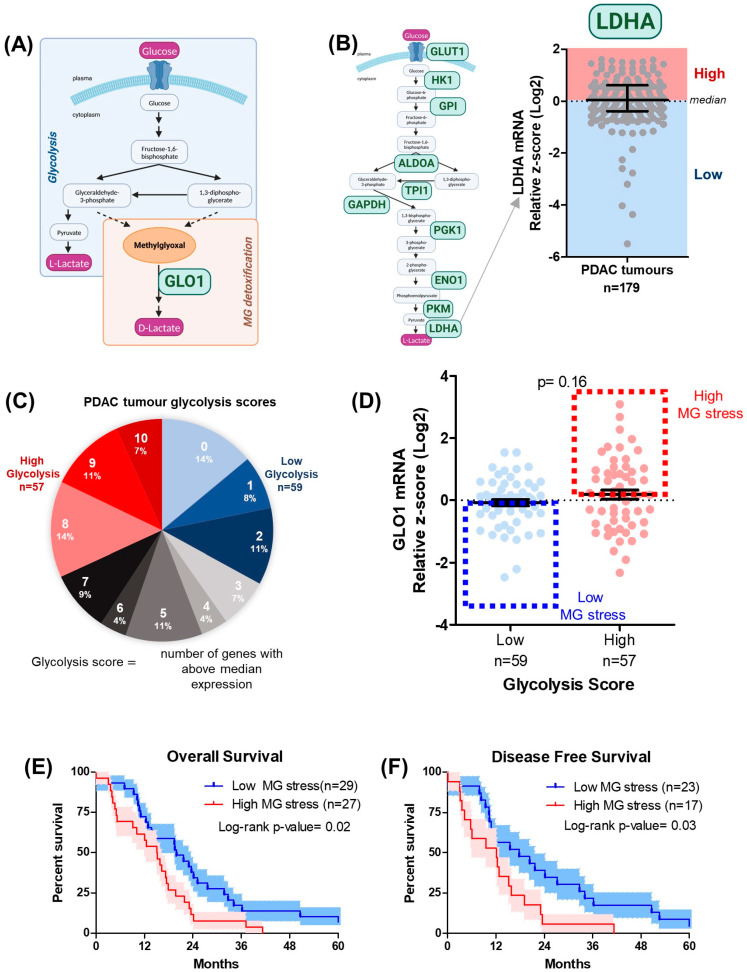
MG stress is associated with poor patient outcome in PDAC. (**A**) Schematic describing spontaneous MG production from glycolytic intermediates and MG detoxification to D-lactate through the glyoxalase system. (**B**) Schematic representing glycolysis enzymes (green) for which expression was split into high and low based on median, for *n* = 179 PDAC tumors (TCGA data). (**C**) Glycolysis scores for each tumor were determined as the number of times that tumor appeared in the high expression group (above median) for each enzyme. Tumors with a score of 0–2 and 8–10 were classed as low and high glycolytic tumors, respectively. (**D**) Glycolysis scores combined with GLO1 expression (TCGA) to identify high (*n* = 27) and low (*n* = 29) MG-stressed tumors. Kaplan–Meier (KM) curves comparing (**E**) overall survival (months) and (**F**) disease-free survival (months) between high and low MG-stressed tumors. Red and blue shading represents SEM and data were analyzed using the Mantel–Cox test.

**Figure 2 cells-12-01414-f002:**
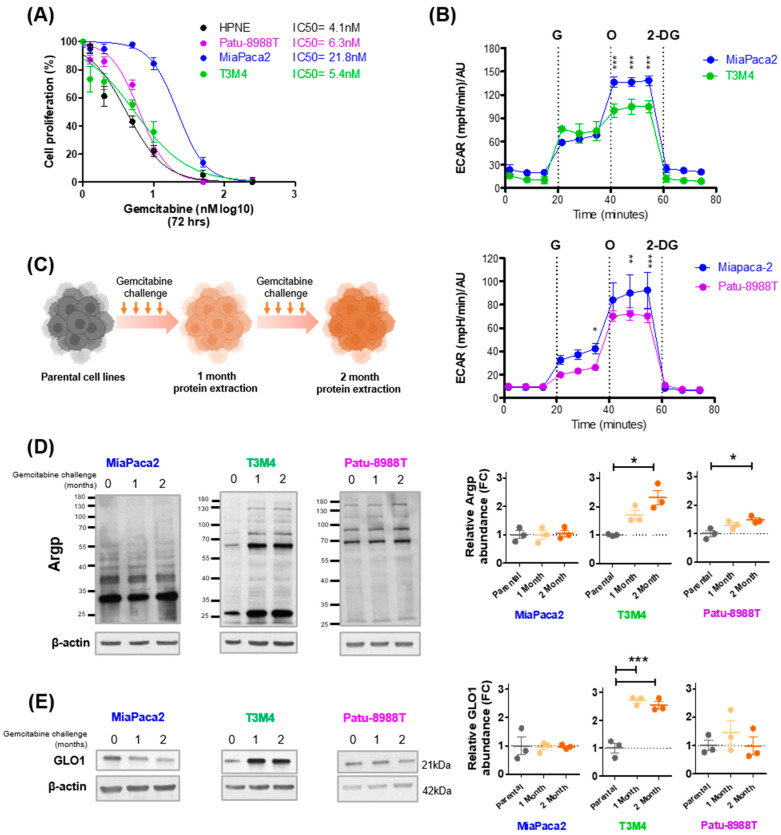
Gemcitabine increases MG stress in gemcitabine-sensitive pancreatic cancer cells. (**A**) Determination of half-maximal inhibitory concentration (IC50) for gemcitabine in Patu-8988T, MiaPaca2, T3M4, and HPNE cells. Cells were treated with gemcitabine for 72 h and proliferation was assessed using Hoechst incorporation. Data were normalized to control and presented as mean ± SEM for three independent experiments. IC50 values were calculated using variable slope modeling. (**B**) Kinetic measurement of extracellular acidification rate (ECAR) in MiaPaca2, T3M4, and Patu-8988T cells in response to glucose (G; 20 mM), oligomycin (O; 1µM), and 2-deoxyglucose (2-DG; 50 mM). ECAR data were normalized with cell number, and evaluated using Hoechst incorporation (arbitrary unit [A.U.]). One representative experiment out of three is shown. Each data point represents mean ± SEM. *** *p* < 0.001. (**C**) Schematic representation for the experimental design of short-term gemcitabine treatments (1 and 2 months) in parental Patu-8988T (50 and 80 nM, respectively), MiaPaca2 (20 and 50 nM, respectively), and T3M4 (50 and 100 nM, respectively) cells. * *p* < 0.05. ** *p* < 0.01. *** *p* < 0.001. Representative Western blot and quantification of (**D**) argpyrimidine (Argp) and (**E**) GLO1, respectively, in Patu-8988T, MiaPaca2, and T3M4 PDAC cell lines treated with gemcitabine, as indicated in (**C**). β-actin was used for normalization, and data were analyzed using one-way ANOVA (Tukey post hoc) and shown as mean values ± SEM of three independent experiments. * *p* < 0.05. ** *p* < 0.01. *** *p* < 0.001.

**Figure 3 cells-12-01414-f003:**
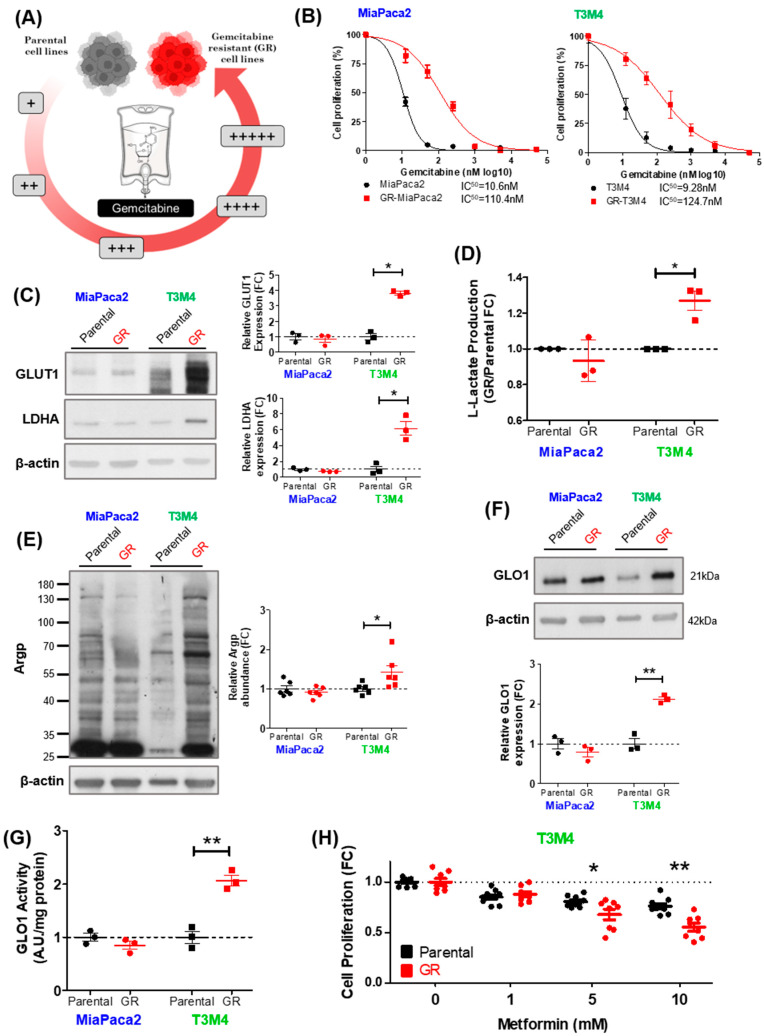
Association between MG stress and gemcitabine resistance in PDAC. (**A**) Schematic representing the generation of gemcitabine-resistant (GR) PDAC cells from parental PDAC cells, using incrementally increasing concentrations of gemcitabine in culture over time. (**B**) Determination of half-maximal inhibitory concentration (IC50) for gemcitabine in parental and GR MiaPaca2 and T3M4 PDAC cells. Cells were treated with gemcitabine for 72 h and proliferation was assessed with Hoechst incorporation. Data were normalized to control and presented as mean ± SEM for three independent experiments. IC50 values were calculated using variable slope modeling. (**C**) Representative Western blot and quantification of GLUT1 and LDHA expression in parental compared to GR MiaPaca2 and T3M4 PDAC cell lines. β-actin was used for normalization, and data were analyzed using a paired t-test and shown as mean values ± SEM of three independent experiments. * *p* < 0.05. (**D**) L-lactate production in 48 h conditioned media from parental and GR MiaPaca2 and T3M4 PDAC cells. Data were analyzed using a paired t-test and shown as mean values ± SEM of three independent experiments. * *p* < 0.05. Representative Western blot and quantification of (**E**) argpyrimidine (Argp) and (**F**) GLO1 expression in parental compared to GR MiaPaca2 and T3M4 PDAC cell lines. β-actin was used for normalization, and data were analyzed using a paired t-test and shown as mean values ± SEM of three independent experiments. * *p* < 0.05. ** *p* < 0.01. (**G**) GLO1 maximal activity, measured in parental and GR MiaPaca2 and T3M4 PDAC cells, is expressed as arbitrary units (A.U) per mg proteins. Data were analyzed using a paired t-test and shown as mean values ± SEM of three independent experiments. ** *p* < 0.01. (**H**) Parental and GR T3M4 cell proliferation assessed using Hoechst incorporation after 24 h metformin (1, 5, and 10 mM) treatment. Data were normalized to untreated cells, and analyzed using a paired t-test and presented as mean ± SEM for three independent experiments. * *p* < 0.05. ** *p* < 0.01.

**Figure 4 cells-12-01414-f004:**
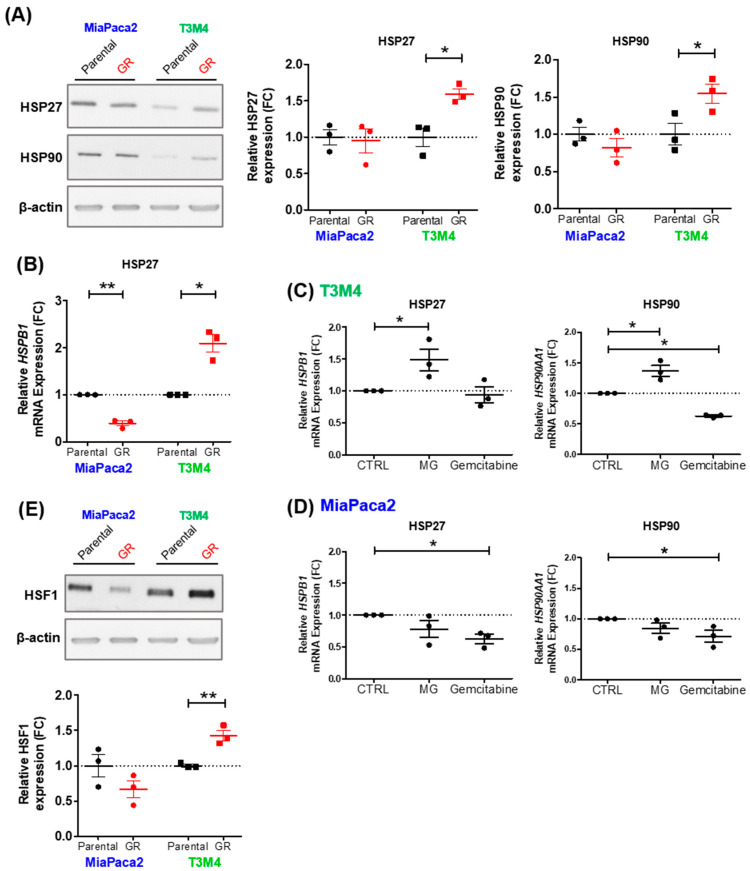
Regulation of heat shock response by MG in gemcitabine-resistant PDAC cells. (**A**) Representative Western blot and quantification of HSP27 and HSP90 expression, respectively, in parental compared to GR MiaPaca2 and T3M4 PDAC cell lines. β-actin was used for normalization, and data were analyzed using a paired t-test and shown as mean values ± SEM of three (or six) independent experiments. * *p* < 0.05. (**B**) mRNA levels of HSP27 (HSPB1) assessed by RT-qPCR in parental compared to GR MiaPaca2 and T3M4 cells. mRNA levels are shown as relative to parental cells and data were analyzed using a paired t-test and shown as mean values ± SEM of three independent experiments; * *p* < 0.05. ** *p* < 0.01. Parental (**C**) T3M4 and (**D**) MiaPaca2 cells were evaluated for mRNA levels of HSP27 (HSPB1) and HSP90 (HSP90AA1) after 24 h treatment with MG (100 µM) or gemcitabine (100µM). mRNA levels are shown as relative to untreated parental cells. Data were analyzed using a paired t-test and shown as mean values ± SEM of three independent experiments. * *p* < 0.05. (**E**) Representative Western blot and quantification of HSF1 expression in parental compared to GR MiaPaca2 and T3M4 PDAC cell lines. β-actin was used for normalization, and data were analyzed using a paired t-test and shown as mean values ± SEM of three independent experiments. ** *p* < 0.01.

**Figure 5 cells-12-01414-f005:**
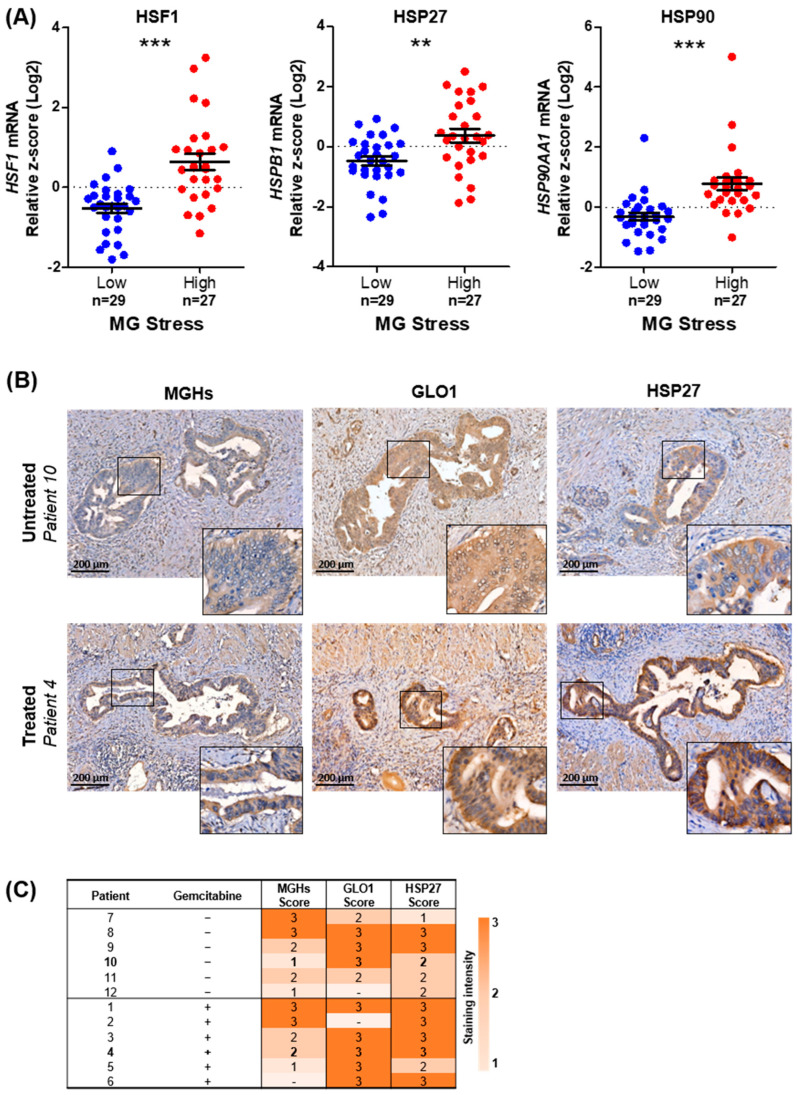
Association between heat shock response, MG stress, and gemcitabine challenge in PDAC tumors (**A**) Comparison of HSF1, HSPB1, and HSP90AA1 mRNA expression between high (*n* = 29) and low (*n* = 27) MG-stressed tumors (TCGA data), as determined in Figure 1B–D. Data were analyzed using unpaired Student’s t-test and shown as mean values ± SEM. ** *p* < 0.01. *** *p* < 0.001. (**B**) Representative images of MGHs, GLO1, and HSP27 staining, as assessed by IHC, in gemcitabine-treated (patient 4) and untreated (patient 10) PDAC tumors (5× magnification). Identical tumor regions are shown where possible. Regions of higher magnification are outlined by black squares. (**C**) Staining intensity of MGHs, GLO1, and HSP27 in a series of PDAC tumors treated with (*n* = 6) or without (*n* = 6) gemcitabine was scored on a scale of 0–3 (0 = no staining; 1 = weak; 2 = moderate; 3 = high; - = not available). Patient tumors selected for representative images are highlighted in bold.

**Figure 6 cells-12-01414-f006:**
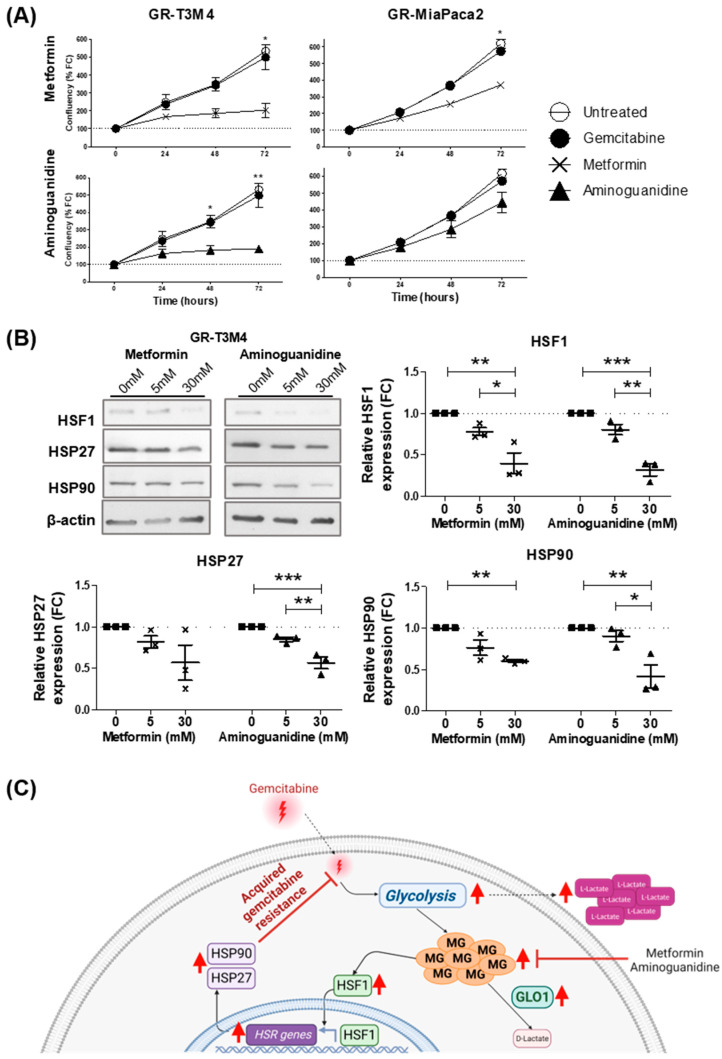
Effect of MG scavenging on gemcitabine resistance and heat shock response in PDAC cells. (**A**) Real-time proliferation of GR-T3M4 and GR-MiaPaca2 cells was assessed using the IncuCyte S3 Live-Cell analyzer, following treatment with either metformin (10 mM; squares), aminoguanidine (10 mM; triangle) or gemcitabine (50 nM; circle). Cell proliferation (%) is shown relative to control (untreated) GR cells. Data were analyzed using an unpaired t-test (48- or 72 h data compared independently) and are shown as mean values ± SEM of two independent experiments. * *p* < 0.05. ** *p* < 0.01. (**B**) Representative Western blot and quantification of HSF1, HSP27, and HSP90 expression, respectively, in GR-T3M4 cells treated with metformin (5 and 30 mM) and aminoguanidine (5 and 30 mM) for 24 h. β-actin was used for normalization, and data were analyzed using a paired t-test and shown as mean values ± SEM of three independent experiments. * *p* < 0.05. ** *p* < 0.01. *** *p* < 0.001. (**C**) Graphical summary depicting the acquisition of gemcitabine resistance in PDAC, whereby gemcitabine-stimulated glycolysis results in methylglyoxal (MG)-mediated heat shock response (HSR) upregulation. Heat shock factor 1 (HSF1); heat shock protein 27 (HSP27); heat shock protein 90 (HSP90); and glyoxalase 1 (GLO1).

**Table 1 cells-12-01414-t001:** Clinicopathological data for PDAC patient tumors.

PDAC Patient	Neo-Adj Chemo	Sex	Age (Diag)	TNM Stage (at Diag)	Total # of Neo-Adj Cycles	Survival since Diag (Months)	Mets (Y/N)	KRAS Variant
1	Gem	F	69.3	IIB	3	5	Yes	G12V
2	Gem	M	65.2	IIB	3	29	Yes	-
3	Gem	M	59	IIB	3	74.9	Yes	-
4	Gem	M	70.2	IB	3	19.9	Yes	-
5	Gem	F	87.7	IIB	4	16.8	-	-
6	Gem/Nab-P	M	71.7	III	3	48.1	No	-
7		F	77.3	IIB	-	7.6	Yes	-
8		F	50	IA	-	97.8	Yes	wt
9		M	76.7	III	-	9.4	Yes	Q61H
10		M	74	IB	-	15	Yes	G12D
11		F	62.9	IA	-	99.2	Yes	Q61H
12		F	82.9	IB	-	36	No	wt

Gem = gemcitabine; diag = diagnosis; wt = wildtype; Mets = metastasis.

## Data Availability

Not applicable.

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
