# Peer review of "Resistance to Gemcitabine in Pancreatic Cancer Is Connected to Methylglyoxal Stress and Heat Shock Response"

_cells, 2023, doi:10.3390/cells12101414_

Round 1
Reviewer 1 Report
The manuscript by Crake et al conducted studies to determine the underlying mechanisms involved in gemcitabine resistance development in PDAC. Initially, they analyzed the published the TCGA database to determine the importance of MG stress on PDAC patient outcome and showed that patients with tumors showing high MG stress show poor overall survival and disease-free survival. To validate this, additional studies were conducted on in vitro cultured PDAC cells. Three different PDAC cell lines along with a normal HPDE cell line was used in the initial studies. Two of the PDAC cells were made resistant to gemcitabine by dose escalation method. Additionally, studies were also conducted on PDAC tumors from patients who were treated or not treated with gemcitabine to determine if they show any differences in MG stress associated molecules. Their results show that MG-mediated activation of heat shock response plays a major role in development of gemcitabine resistance and scavenging MG using metformin or aminoguanidine enhances the sensitivity of PDAC cells, especially the gemcitabine-resistant cells, to the gemcitabine treatment. Overall, the manuscript is well written, with minor grammatical errors, and the data supports the conclusion. A few major and minor points are provided below for the authors:
Major Points:
1. The authors compared the IC50 values for gemcitabine on HPNE and the 3 PDAC lines initially but HPNE was not used in any of the additional studies. Does treatment of HPNE with gemcitabine show alterations in Argp, GLO1 or any of the HSR associated proteins, like that is presented with MiaPaca2, T3M4 or Patu-8988T? Is the effect specific to PDAC cells?
2. How was the concentration of gemcitabine selected for the results shown in Figure 2D and E? According to the authors’ MiaPaCa2 is more resistant to gemcitabine than the other two cell lines, but the concentration used to test Argp and GLO1 is lower for MiaPaCa2 than the other two lines, would that be a reason for the altered effect? Also, the effect of gemcitabine on Patu-8988T is very weak or none for both Argp and GLO1, why are these cells showing differential effects? Any discussion on this would be helpful to the readers.
3. Figure 2A and 3B have data showing IC50 analysis for MiaPaCa2 and T3M4; according to the authors T3M4 is more sensitive to gemcitabine than MiaPaCa2 but the data in Figure 3B does not agree with this statement, both MiaPaCa2 and T3M4 parent cells show almost similar IC50, what is the reason for this discrepancy, please provide an explanation.
Minor:
1. Manuscript needs to be edited for grammatical errors.
2. Line 171-172, The authors mention ‘to block non-specific serum-binding sites’, this can be changed to ‘to block non-specific binding’.
Overall the manuscript is well written and needs some minor edits, especially for grammatical errors.
Author Response
"Please see the attachment."

Reviewer 2 Report
Crake and colleagues aimed to analyze the link between gemcitabine resistance, heat shock response (HSR) and methylglyoxal (MG) stress in glycolytic pancreatic ductal carcinoma (PDAC) tumors. They found that MG stress is detected, more often, in gemcitabine-treated tumors, when compared to untreated ones and it is also associated with worse patient outcome. In PDAC models of acquired gemcitabine resistance, they find that MG favors cell survival and gemcitabine tolerance through the regulation of HSR. This novel finding allows them hypothesize that the use of MG scavengers, such as metformin and aminoguanidine could prevent the consequences of glycolytic rewiring, in PDAC tumors of patients under gemcitabine therapy.
The premise of the study is interesting and the paper has a potential to be accepted, although some points need to be clarified or fixed before a positive action can be taken considering it for publication.
I would like the authors to consider the following suggested minor revisions
1.- I think it could be better to say “pancreatic ductal adenocarcinoma”, instead of “pancreatic ductal carcinoma” when using the initials “PDAC”.
2.- Fig. 5B: Could you provide higher magnification images of MGHs, GLO1 and HSP27 immunohistochemical staining, in treated and untreated tumors? Staining intensity is difficult to analyze with 5X magnification.
3.- Could the authors provide the pattern of staining for each of the antibodies used? Is it nuclear, cytoplasmatic, membranous?
4.- How many fields and/or cells do the authors study (under the microscope) of each tumor sample to evaluate immunohistochemical antibodies?
After these problems are fixed, the paper can be reconsidered for publication.
The manuscript presents few grammatical mistakes and typographical errors that could improve following a revision by a native speaker.
Author Response
"Please see the attachment."
